# Machine learning uncovers independently regulated modules in the *Bacillus subtilis* transcriptome

Kevin Rychel [1], Anand V. Sastry [1] & Bernhard O. Palsson [1,2,3] ✉

The transcriptional regulatory network (TRN) of *Bacillus subtilis* coordinates cellular functions of fundamental interest, including metabolism, biofilm formation, and sporulation. Here, we use unsupervised machine learning to modularize the transcriptome and quantitatively describe regulatory activity under diverse conditions, creating an unbiased summary of gene expression. We obtain 83 independently modulated gene sets that explain most of the variance in expression and demonstrate that 76% of them represent the effects of known regulators. The TRN structure and its condition-dependent activity uncover putative or recently discovered roles for at least five regulons, such as a relationship between histidine utilization and quorum sensing. The TRN also facilitates quantification of population-level sporulation states. As this TRN covers the majority of the transcriptome and concisely characterizes the global expression state, it could inform research on nearly every aspect of transcriptional regulation in *B. subtilis*.

[1] Department of Bioengineering, University of California San Diego, La Jolla, CA 92093, USA. [2] Department of Pediatrics, University of California San Diego, La Jolla, CA 92093, USA. [3] Novo Nordisk Foundation Center for Biosustainability, 2800 Kongens Lyngby, Denmark. ✉email: palsson@ucsd.edu

Cells interpret dynamic environmental signals to govern gene expression through a complex transcriptional regulatory network (TRN). *Bacillus subtilis*, a model gram-positive soil and gut bacterium, is one of the most widely studied species in microbiology, providing a rich background for understanding its TRN. This generalist organism is a model for processes such as sporulation[1], biofilm formation[2], and competence[3]—all of which are key to understanding pathogenesis in other bacteria, such as *Staphylococcus aureus* and *Clostridium difficile*. *B. subtilis* is also commonly engineered for industrial production purposes[4], which creates demand for practical knowledge about how it responds to stimuli and alters its gene expression.

In 2012, Nicolas et al.[5], generated a transcriptomic microarray data set of *B. subtilis* with 269 expression profiles under 104 conditions, which included growth over time in various media, carbon source transitions, biofilms, swarming, various nutritional supplements, a variety of stressors, and a time course for sporulation. The wide scope and high quality of this data set have led to its broad adoption. It is now the expression compendium featured on *Subti*Wiki, an online resource for *B. subtilis* that is one of the most widely used and complete databases for any organism[6]. *Subti*Wiki contains detailed biological descriptions and binding sites for hundreds of transcriptional regulators; however, binding sites alone cannot explain the condition-specific transcriptomic responses of bacteria to dynamic environmental conditions[7,8].

Independent component analysis (ICA) is an unsupervised statistical learning algorithm that was developed to isolate statistically independent voices from a collection of mixed signals[9]. ICA applied to transcriptomic matrices simultaneously computes independently modulated sets of genes (termed iModulons) and their corresponding activity levels in each experimental condition[10]. iModulons can be interpreted as data-driven regulons, though they rely on observed expression changes instead of transcription factor binding sites. The condition-dependent activity level of iModulons indicates how active the underlying regulator is. Since the number of iModulons is substantially fewer than the number of genes, they are a significantly easier way to analyze systems-level cell behavior.

ICA has been shown to extract biologically relevant transcriptional modules for a variety of transcriptomic datasets, especially in yeast and human cancer[11–15]. It was the best out of 42 methods at recovering known co-regulated gene modules in a comprehensive examination of TRN inference methods[16]. ICA also obtained the most robust modules across datasets compared to similar factorization algorithms[17]. We previously applied this approach to a large, high-quality *Escherichia coli* RNA-seq compendium and extracted 92 iModulons, two-thirds of which exhibited high overlap with known regulons[10]. This analysis provided many insights into the *E. coli* TRN, including the addition of genes to known regulons (validated through ChIP-exo), bifurcation of the purine synthesis regulon, the characterization of new regulons, and identification of clear associations between regulator mutations and activities. We have also applied ICA to transcriptomics of evolved strains to understand evolutionary trade-offs and regulatory adaptations in naphthoquinone-based aerobic respiration[18], and to characterize the function of the transcription factor OxyR, which responds to peroxide[19].

Without using ICA, others have attempted to infer the TRN of *B. subtilis*. Arietta-Ortiz et al.[20] used an "Infereletor" approach which utilized prior knowledge of the TRN along with transcriptomics (including the Nicolas et al. data) to obtain a global network, infer activity levels, and predict new TF-gene interactions. In addition, Fadda et al.[21] used genomic regulatory motifs of major regulators to infer a TRN, and Leyn et al.[22] combined a variety of available data types to infer regulons in *B. subtilis* as well as 10 related *Bacillales* species. These approaches have been valuable for expanding our understanding of the TRN and can be especially helpful in complex processes like sporulation where transcriptomics can be supplemented with other data types. However, prior methods suffer from a bias toward the known aspects of the TRN, which can pose a barrier for new discovery or unbiased validation of past data. They are also not as easily applicable to organisms with very incomplete TRN annotations. This motivates the development of fully unsupervised approaches like ICA.

Given our success with ICA applied to RNA-seq data from a model gram-negative bacterium, we sought to determine what it can uncover about a microarray data set from a model gram-positive bacterium. Though RNA-seq data exists for *B. subtilis*, the Nicolas et al. data set has a comparatively wider diversity of conditions and a more established reputation for data quality. We have shown that the condition space is more important than the technology used[10,23], which makes this a good choice of data set. Using the wealth of TRN knowledge available on *Subti*Wiki, this analysis uncovers many insights. We determine the main functions and regulators that control a large fraction of the transcriptome, and we characterize the iModulon accuracy in relation to the known TRN. iModulon activities reveal relationships and stimuli that have been present in the data but never specifically investigated; it is therefore a powerful hypothesis-generating tool. We specifically present five unexpected iModulon activations and hypotheses about their mechanisms. We characterize sporulation, which led us to the identification of three major transcriptomic stages in the process, including iModulons for the known sigma factor cascade. Finally, we present three transcriptional units with a little prior characterization that warrant further study.

## Results

**Independent component analysis reveals the structure of the *B. subtilis* transcriptome.** We performed ICA on the Nicolas et al.[5], data set (see "Methods" section, Supplementary Data 1 and 2) and obtained 83 robust iModulons (Supplementary Data 3–6). These 83 iModulons constitute the statistically independent gene expression signals found across the conditions used in the generation of this data. Together, they contain 36.25% of the genome and explain 72% of the variance in gene expression (Supplementary Methods, Supplementary Fig. 1b). The distribution of the number of genes in each iModulon follows a power law, similar to the power law for the connectivity of TFs in literature regulatory networks[24,25] (Supplementary Fig. 2a, b).

Unlike regulons, which are sets of co-regulated genes based on a variety of experimental results in the literature, iModulons are derived solely from the measured transcriptome through an unbiased method (Fig. 1a). However, the known regulon structure of the TRN is largely recapitulated by the iModulons. 63 of the 83 iModulons were successfully mapped to a known regulator, and an additional 3 are likely to be co-regulated by unknown mechanisms. The iModulon-derived TRN covers 2235 gene/iModulon relationships, of which 1536 are known gene/regulator interactions and 699 are new (Supplementary Data 8). Our TRN structure contained seven iModulons that exhibited perfect overlap with annotated regulons and whose activity levels match expectations, such as MalR (Supplementary Note 1, Supplementary Fig. 3). This illustrates that independent signals such as transcription factor binding, which dictate gene expression, lead to observable signals in the TRN from condition to condition, and ICA was able to identify them. Graphical summaries of all iModulons, including their gene sets, activities, overlap with regulons, and upstream motifs (Supplementary Note

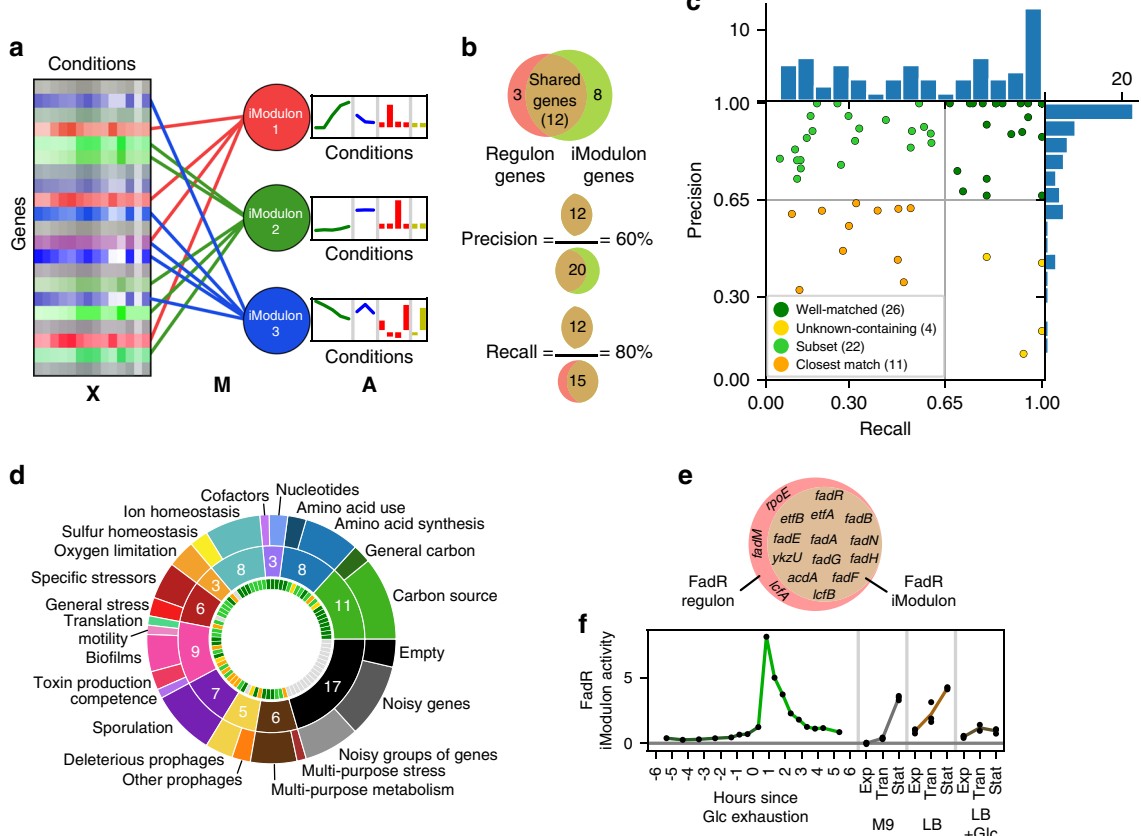

**Fig. 1 Independent component analysis (ICA) extracts regulatory signals from a compendium of transcriptomic data. a** Given a matrix of gene expression data, **X** (Supplementary Data 2), ICA identifies independently modulated sets of genes (iModulons) in the transcriptome which are linked to genes through the matrix **M** (Supplementary Data 3). Three iModulons are symbolically represented; the red iModulon consists of four genes, and the green and blue iModulons consist of five genes. The condition-dependent activities of the iModulons are stored in matrix **A** (Supplementary Data 4). The bar chart indicates the activity levels of the iModulons under different conditions, where the colors indicate different experiments. The three matrices are related as **X** = **M**\***A**. **b** Graphical representation of the definitions of precision and recall of a given iModulon and the corresponding regulon (example numbers are shown). **c** Scatter plot of precision and recall of the enrichments for the 63 (out of 83) iModulons that were matched to a regulon. Histograms in the margins demonstrate the high precision of most enrichments (see Supplementary Data 7, Supplementary Fig. 1c for more details). **d** Donut chart of iModulon functions. The outermost ring lists specific functions and the center ring lists broad functions, with the number of iModulons in the broad category shown in white. The innermost ring shows the regulon confidence quadrant of the corresponding iModulon, as defined in **c**. **e**, **f** An example iModulon that was enriched for FadR. **e** Venn diagram of the FadR iModulon genes and the FadR regulon (non-coding RNAs have been omitted). **f** Activity level found in a row of A for four experiments (separated by vertical gray lines) from the data set. Activity levels increase during growth in the absence of glucose (M9 media, gray; LB media, light brown), remain low during growth in the presence of glucose (dark green, dark brown), and spike upon glucose (Glc) starvation (green). "Exp", "Tran" and "Stat" refer to exponential, transition, and stationary phase, respectively. See Supplementary Data 1 for detailed growth conditions.

7, Supplementary Data 10) are presented in Supplementary Data 6 and online at imodulondb.org[26].

iModulons are given a short name, usually based on their enriched regulator. If multiple regulators control an iModulon, their names are separated by "+" to indicate the intersection of the regulons, or "/" to indicate the union of the regulons. In some cases, a different name was chosen based on the primary regulator, gene prefix, or most representative gene in the set (Supplementary Data 7).

The relationship between iModulons and regulators can be characterized by two measures: (1) precision (the fraction of iModulon genes captured by the enriched regulon) and (2) recall (the fraction of the regulon contained in the iModulon) (Fig. 1b). These two measures can be used to classify iModulons into six groups (Fig. 1c). (1) The well-matched group ($n = 26$) has precision and recall greater than 0.65. It includes several regulons with local regulators that are associated with specific metabolites. (2) The subset iModulons ($n = 22$) exhibit high precision and low recall. They contain only part of their enriched regulon, perhaps because the regulon is very large and only the genes with the most transcriptional changes are captured. This group contains global metabolic regulators such as CcpA and CodY, as well as the stress sigma factors. (3) A third group, deemed unknown-containing ($n = 4$), has low precision but high recall. These iModulons contain some co-regulated genes along with unannotated genes which may have as-yet-undiscovered relationships to the enriched regulators (Supplementary Data 8), or at least be co-stimulated by the conditions in the data set. (4) The remaining enriched iModulons are called the closest match ($n = 11$) because neither their precision nor recall met the cutoff, but the grouping had statistically significant enrichment levels and appropriate activity profiles. The difference in gene membership between these iModulons and their regulons provide excellent targets for discovery. The iModulons with no enrichments comprise the last two groups: (5) new regulons ($n = 3$) are likely to be real regulons with unexplored transcriptional mechanisms, while (6) the

remaining uncharacterized iModulons were likely to be noise due to large variance within conditions or the fact that they contain one or fewer genes.

Functional categorization of iModulons provides a systems-level perspective on the transcriptome (Fig. 1d). Metabolic needs account for approximately one-third of the iModulons, while comparatively fewer iModulons deal with stressors, lifestyle choices such as biofilm formation and sporulation, and mobile genetic elements like prophages. Some iModulons have multiple biological functions, such as one which synthesizes both nicotinamide and biotin. These iModulons may result from co-stimulation of the different functions by all conditions probed in the data set (e.g., both nicotinamide and biotin synthesis were always stimulated together by minimal media, so the algorithm could not separate them into unique signals).

The FadR iModulon provides an example of the information encoded by the iModulon gene membership (Fig. 1e) and activities (Fig. 1f). All genes within this iModulon are regulated by FadR, so this enrichment has 100% precision. Three genes that are annotated as belonging to the FadR regulon were not captured in the iModulon—*lcfA*, *rpoE*, and *fadM*. However, all three have additional regulation separate from that of FadR[27,28], which may lead them to have a divergent expression from the rest of the iModulon. The activity levels (Fig. 1f) reflect expectations: FadR genes are repressed by FadR in the presence of long-chain acyl-coA, and FadR itself is repressed by CcpA in the presence of fructose-1,6-bisphosphate[28], which causes the expression to rise as nutrients (specifically sugars and fats) are depleted, and to be particularly strong immediately following glucose exhaustion. As this example illustrates, the precision and recall are sensitive to developments in regulon annotations; they improve as regulon annotations become more complete (Supplementary Note 8)[29].

**iModulons generate hypotheses**. iModulon activities can often be explained by prior knowledge, as was the case with FadR. However, they can also present surprising relationships that lead to the generation of hypotheses or strengthen arguments for recently proposed mechanisms. In the subsequent sections, we list five such examples, and more are provided in the Supplementary Notes (Supplementary Notes 3–5).

**Ethanol may stimulate tryptophan synthesis**. The tryptophan synthesis iModulon (*trpEDCFB*) was strongly activated under ethanol stress (Fig. 2a), a response that has not been previously documented in bacteria. This iModulon is regulated by the *trp* attenuation protein (TRAP), which represses its genes in the presence of tryptophan[30]. Therefore, this activation indicates that ethanol is probably depleting intracellular tryptophan concentrations. Exploring the tryptophan synthesis pathway reveals a hypothetical mechanism for this depletion: flux from the precursor chorismate may be redirected to replenish folate that has been damaged by ethanol oxidation byproducts[31] (Supplementary Fig. 5a). If this hypothesis is accurate, it may inform research on the tryptophan deficiency and neurotransmitter metabolism problems observed in human alcoholic patients[32,33], especially given that *B. subtilis* is an important folate producer in the gut microbiome[34,35].

**Histidine may be utilized by quorums**. The HutP iModulon for histidine utilization (*hutHUIGM*) is controlled by an anti-terminator that derepresses it in the presence of excess histidine, as well as by the master regulators CcpA and CodY; therefore, its activation indicates that histidine is plentiful while other amino acids are not and that carbon sources are poor[36]. Surprisingly, it was by far most strongly activated in confluent biofilms and

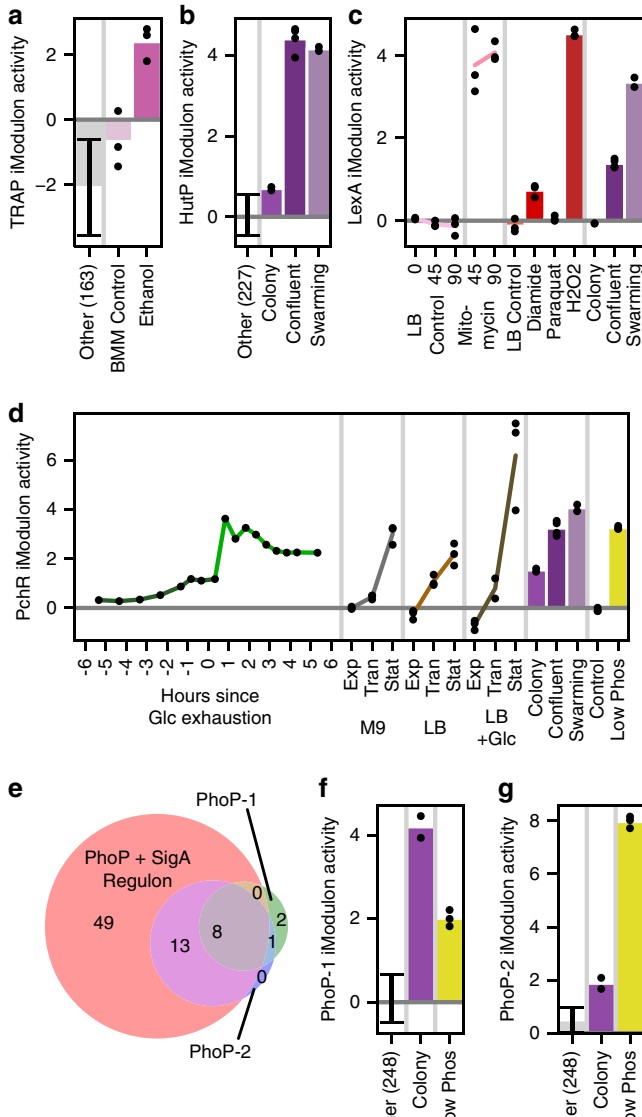

**Fig. 2 iModulons provide a range of insights.** Error bars: mean ± standard deviation; black dots indicate separate samples; vertical gray lines separate different experiments in the data set. Unless otherwise stated, "Other" category includes all conditions except sporulation and those shown, with the number of samples included in parentheses. **a** Tryptophan synthesis (TRAP) iModulon activity, which is unexpectedly elevated by ethanol (Supplementary Fig. 5a). The experiment was carried out in Belitsky minimal medium (BMM). The "Other" category excludes carbon source transition experiments, in which this iModulon exhibits technical noise. **b** Histidine utilization (HutP) iModulon activity, which is strongest in quorum conditions. **c** LexA iModulon activity is elevated by DNA damage (mitomycin and peroxide) and in swarming (Supplementary Fig. 5b). **d** Pulcherrimin (PchR) iModulon activity increases when growth is expected to slow, especially in the stationary phase in rich (LB) media containing glucose (Glc). "Exp", "Tran" and "Stat" refer to exponential, transition, and stationary phase, respectively. **e** Venn diagram of gene presence in the PhoP+SigA regulon and related iModulons. Numbers indicate the amount of genes or non-coding RNAs in each subset. Although the iModulons are significantly enriched for the intersection of the PhoP and SigA regulons, they have been named PhoP-1 and PhoP-2 for simplicity. **f, g** Bar graphs of PhoP iModulon activity demonstrating the use of PhoP-1 for early biofilm growth ("Colony" refers to individual colonies on a plate after 16 h) and PhoP-2 for extreme phosphate starvation ("Low Phos" indicates phosphate starvation for 3 h).

swarming cells (Fig. 2b). Independent colonies from the same experiment do not exhibit activation, which leads us to rule out the media composition as the reason for these activity levels. The connection between these lifestyle conditions and histidine metabolism has not been studied in *B. subtilis*, but it has been observed in *A. baumannii*, where histidine degradation was shown to be upregulated in proteomic studies of biofilms, and histidine supplementation stimulated increased biofilm production[37]. Two recent studies discovered that biofilm-inhibiting antimicrobials worked by suppressing histidine synthesis in *Staphylococcus xylosus*[38,39]. One proposed mechanism implicated the production of extracellular DNA, which is an important component of both *A. baumannii* and *B. subtilis* biofilms[40]. Given that this iModulon is also activated by swarming cells, an alternative hypothesis may be that HutP is involved with quorum sensing or surfactant production: both activating conditions have a quorum and high surfactant production, while independent colonies do not.

**DNA damage may stimulate swarming**. The LexA iModulon regulates the SOS response for DNA protection and repair. It is strongly activated by three conditions (Fig. 2c). LexA stimulation by mitomycin and hydrogen peroxide is expected since those conditions damage DNA[41,42]. Unexpectedly, this iModulon is also activated in swarming cells despite a lack of DNA damaging agents in that condition. We propose a potential mechanism for this activation: recent research has indicated that certain cells in a culture will tend to accumulate reactive oxygen species and DNA damage. Those cells will produce Sda (a developmental checkpoint protein) and form a subpopulation separate from those that produce biofilm[43]. The LexA+, biofilm− population would no longer be producing EpsE, which catalyzes a step in the biofilm synthesis process and also suppresses swarming[44]. In addition, this connection may be mediated by interactions between RecA and CheW, which have been observed in *Salmonella enterica*[45]. Therefore, we predict that DNA damage encourages swarming motility based on iModulon activation and this mechanism (Supplementary Fig. 5b).

**An iron chelator may signal the stationary phase**. The PchR iModulon produces, extrudes, and imports pulcherrimin, an iron chelator[46]. Over all of the exponential to stationary phase growth experiments, we observe increases in PchR activation (Fig. 2d). We also see PchR activation in late-stage biofilm, glucose exhaustion, and phosphate starvation experiments. These results agree with a recent study that found pulcherrimin to be an important intercellular signal for the stationary phase that also helps exclude competing bacteria from established biofilms[47]. The regulation mechanisms of iModulons like this one can be the subject of future research.

**Phosphate limitation stimulates tiers of regulation**. The PhoP regulon controls phosphate homeostasis. It appears as two separate iModulons (Fig. 2e–g). PhoP-1 encodes high-affinity phosphate uptake transporters. Phosphate is used to produce (and is effectively stored in) teichoic acid, which is a major component of the cell wall. As a colony grows, it must uptake phosphate to produce more cell walls—indeed, teichoic acid intermediates are the major stimulus for PhoP activity[48]. It is therefore unsurprising that PhoP-1 is strongly activated in independent colonies, which are exponentially growing in close quarters with low local free phosphate concentrations. PhoP-2 contains PhoP-1 as well as 13 other genes which encode more extreme phosphate recovery strategies: *phoABD*, which salvages phosphate monoesters but produces reactive alcohols, *glpQ*,

which degrades extracellular teichoic acid, and *tuaBCDEFGH*, which replaces teichoic acid with phosphate-free teichuronic acid. PhoP-2 is only active under phosphate starvation, consistent with the extreme strategy it encodes. Perhaps the affinities of the promoters of the PhoP-2 specific genes are lower than that of the PhoP-1 genes, which could lead to this graded response.

**Six iModulons capture the major transcriptional steps of sporulation**. The data set we analyzed contained an eight-hour sporulation time course, which yielded six major sporulation iModulons that were activated sequentially over the first 6 h (Fig. 3a). The identification of these gene sets by ICA indicates coherent expression across the transcriptome, and more dramatic transcriptional variation compared to excluded genes. The conclusions drawn from these iModulons are limited by the complexity of sporulation[1,49] and the stochasticity of its onset[50]. Because of this, we observe many genes shared between consecutive iModulons (Supplementary Fig. 8a). Nonetheless, the following analysis demonstrates that they still provide valuable information, including identifying 20 uncharacterized proteins whose annotations did not previously reflect a putative relationship to sporulation (Supplementary Fig. 8b, Supplementary Data 11).

The gene sets and regulators of the sporulation iModulons roughly match the known sporulation progression (Supplementary Fig. 8e–h). The Spo0A iModulon contains mostly genes known to be activated by high levels of Spo0A~P, including the sigma factors for upcoming sporulation steps, chromosome preparation machinery, and septal wall formation. It is rapidly activated between hours 1 and 2 of the time course. Next, the SigE iModulon carries out functions in the mother cell for engulfment of the forespore. After SigE, a dual SigE/G iModulon is activated, which regulates early spore coat formation by both the mother and forespore cells. The SigG iModulon follows; it contains germination receptors, metabolic enzymes, and stress resistance genes. Finally, the SigK regulon is split into two iModulons with functions including coat maturation and mother cell lysis. The difference between the two SigK iModulons may partially be explained by the action of the TF GerE, which represses members of SigK-1 and activates a large fraction of SigK-2 (Supplementary Fig. 8c, d). This is consistent with the known temporal regulation of the SigK regulon[51]. Notably, SigF is the only absent sigma factor; we believe it was not identified because its genes are expressed simultaneously with the SigE, SigG, and SigE/G iModulons, and because many SigF genes are also under SigG control[52]. Nonetheless, these functions and regulators largely match expectations based on literature, providing an a priori validation of the set of known sporulation steps.

The activity levels of the sporulation iModulons can be viewed as markers of progress through sporulation: high Spo0A activity indicates that new spores are forming, and high SigK-2 indicates that some spores are completing the process. Therefore, we can understand how far along other conditions are based on their sporulation activity levels (Fig. 3b–e). Most conditions have a very low level of activation, but the "glutamate + succinate" and pyruvate supplements to minimal media conditions both have elevated expression across all sporulation iModulons, which indicates that the poor carbon sources in these conditions stimulated sporulation (Fig. 3c). Indeed, pyruvate has been shown to regulate sporulation[53,54]. Some other conditions appear to have made it partway through the process: confluent biofilms, the stationary phase in minimal media, and growth at cold temperature all reached the third of six steps. This is appropriate for these conditions based on previous studies[55–57] (Fig. 3d).

With one exception, the progression from one sporulation iModulon to the next is cumulative: we do not see strong

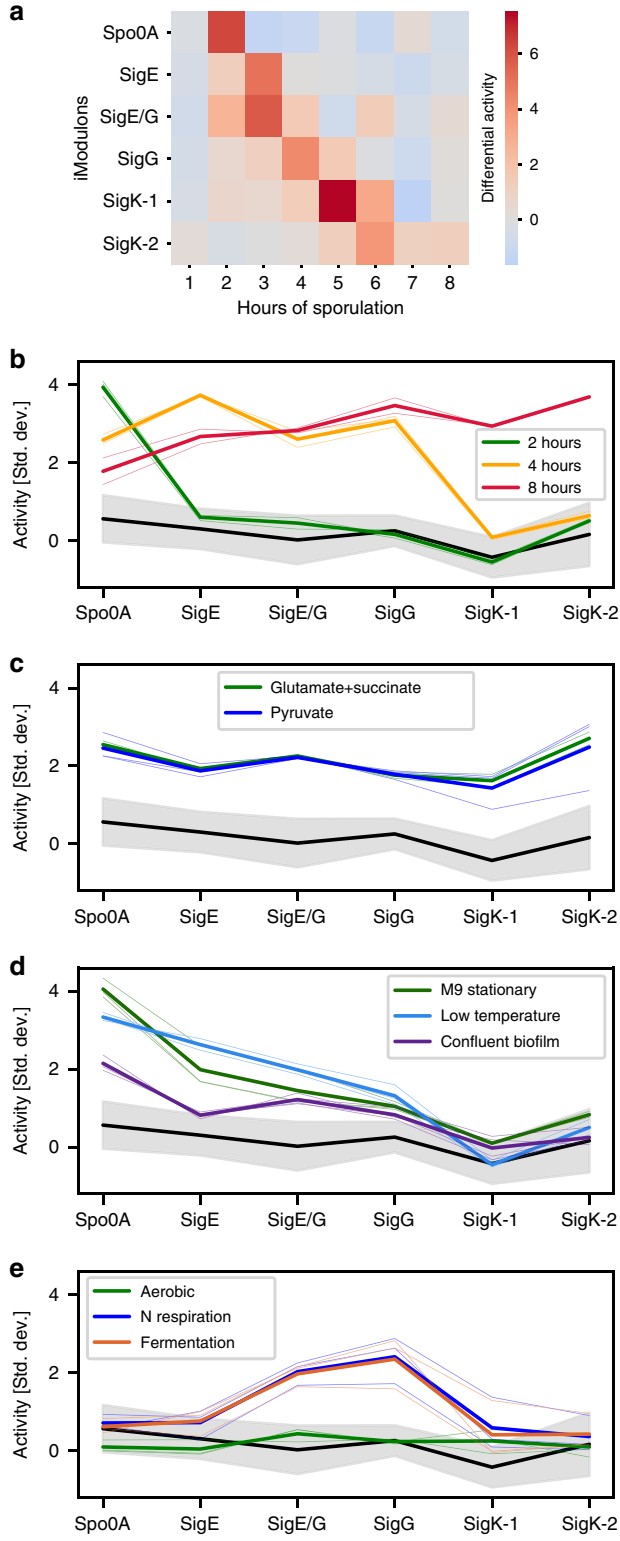

**Fig. 3 Six iModulons (named for their enriched regulators) mark progress through sporulation. a** Heatmap color indicates the change in iModulon activity over the previous hour. **b–e** Line plots of the sporulation progression for selected conditions, with thick lines indicating mean activity and thin lines indicating individual samples. Activity levels were divided by the standard deviation. The black line surrounded by a shaded gray region is the average of all conditions not shown in any plot ± standard deviation ($n = 200$ samples). **b** Three time points of sporulation, showing Spo0A activation at sporulation onset (2 h, green), cumulative expression up to the fourth step (SigG) for an intermediate time point (4 h, orange), and expression of all stages at 8 h (red). **c** Minimal media supplemented with these carbon sources leads to expression of all sporulation iModulons. **d** Three conditions reached the intermediate steps of sporulation. **e** Anaerobic conditions exhibit unusual activity. "Aerobic" is the control condition.

ectopic activation of SigG is limited by negative feedback[60,61] and unlikely to occur in vegetative cells[62]. We, therefore, propose further experiments to determine the role of SigG-dependent genes in anaerobiosis.

**Changes in iModulon activity reveal global transcriptional shifts during sporulation.** In complex processes such as sporulation, the entire cellular transcriptome undergoes system-wide changes beyond those directly related to the process at hand. While much effort has been put into understanding metabolic changes at the onset of sporulation[1,56,58], metabolic, and lifestyle-related regulatory activity are difficult to summarize concisely with previous methods. Because ICA provides a simple method for tracking transcriptome-wide changes, we analyzed activity level fluctuations for the sporulation time course (Fig. 4). Three major stages are involved: a self-preserving metabolic response to amino acid starvation in the first hour, a community-wide lifestyle reallocation in the second hour, and progression through sporulation in the remaining time points.

In the first hour, many amino acid synthesis iModulons (tryptophan, cysteine, arginine, leucine, and threonine) and one amino acid utilization iModulon (histidine) are rapidly activated. This is likely the result of amino acid starvation by the sporulation media, which derepresses these iModulons through transcription factors including CodY. CodY also derepresses the fructosamine consumption iModulon[63] at this time. The AbrB iModulon is derepressed; it responds to nutrient limitation through a variety of functions, including cannibalism[64], that herald the stationary phase and prolong entry into sporulation.

In the second hour, Spo0A is strongly activated in a process that has been widely studied; this marks the onset of sporulation[65]. Also, the histidine utilization of the first hour is compensated by histidine synthesis in the second hour. Zinc, an important cofactor for sporulation proteins[66,67], is taken up. Various colony, biofilm, and antimicrobial iModulons are activated to support the forming spores (DegU, ComA, Eps, Alb). ComK, the competence iModulon, is expressed as an alternative response to starvation. ComK's brief activation at this time point is consistent with the short competence window observed before commitment to sporulation[3]. We also observe the activation of resD, which is typically associated with anaerobic conditions[68,69], and Rex, which regulates overflow metabolism, providing interesting connections to the potential anaerobic activity of SigG discussed in the previous section.

As sporulation continues, fewer non-sporulation iModulons are activated. The notable exceptions are AcoR and FruR, which are both activated around the fourth hour. Both acetoin and

activation of step 2 unless step 1 is active, and so on. This agrees with prior observations[58]. The only exception to this rule is elevated SigG activity by cells in anaerobic conditions (Fig. 3e). This connection is also evident from gene presence: a flavohemoglobin required for anaerobic growth, hmp, is part of the iModulon despite no known connection to SigG. Previous studies have also acknowledged that some SigG-dependent genes are required for anaerobic survival[59]. However, it is known that

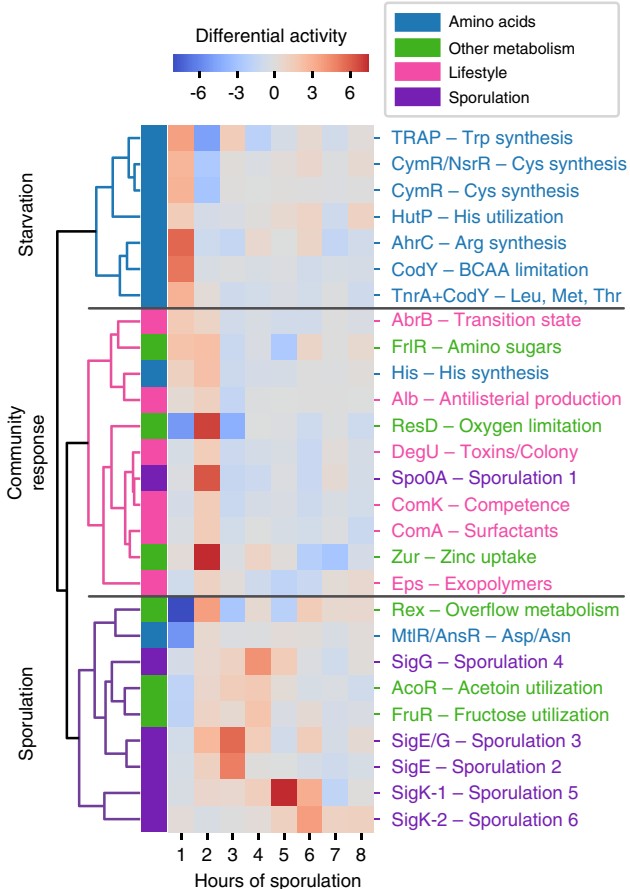

**Fig. 4 Changes in iModulon activity reveal global transcriptional reallocation during sporulation.** Heatmap color indicates a change in iModulon activity over the previous hour. Selected iModulons were hierarchically clustered according to the Pearson *R* correlation between sporulation activity derivatives.

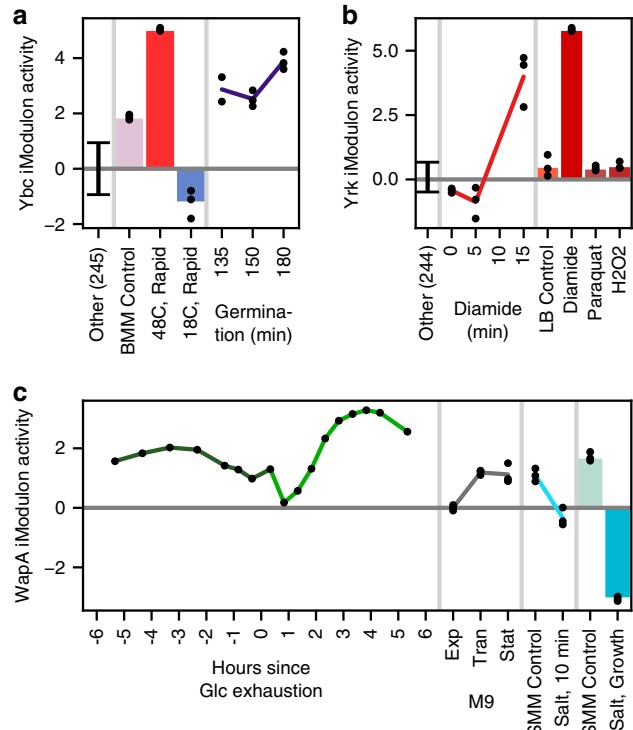

**Fig. 5 The activity levels of uncharacterized iModulons agree with their putative functions.** Bars and lines indicate means, black dots indicate individual samples, and error bars indicate one standard deviation. The "Other" category includes all conditions except the ones in the plot, with the number of samples included in parentheses. Vertical gray lines separate different experiments in the data set. **a** The activity levels of the Ybc iModulon indicate that it may be a response to heat shock or germination. The Belitsky minimal media (BMM) control occurs at 37 °C. **b** The activity levels of the Yrk iModulon (putative sulfur carriers) suggest that it is a response to diamide. The three conditions on the right were taken from LB cultures 10 min after exposure to the labeled stressor. **c** The activity levels of the WapA iModulon indicate activation by nutrient limitation (glucose exhaustion and the three growth phases of M9 media) and suppression by osmotic stress, both in the short (light blue time course) and long term (bars). "Exp", "Tran", and "Stat" refer to exponential, transition, and stationary phase, respectively.

polymeric fructose function as extracellular energy stores[70,71], so perhaps they are used at this stage to provide a final energy source for the completion of sporulation. Overall, these results demonstrate an application of ICA for observing transcriptome-wide changes and lay out the major population dynamics and metabolic changes that underscore spore formation.

**Some poorly characterized iModulons may perform important functions**. Given the vast number of uncharacterized genes in bacterial genomes, ICA can help to narrow the search for new and important regulons by identifying groups of genes with transcriptional co-regulation (Supplementary Data 5, Supplementary Data 8) and their corresponding activity levels. We have identified three iModulons that warrant further study. The first, the *ndhF-ybcCFHI* operon, may be involved in heat shock and germination (Fig. 5a, Supplementary Note 6). Another, the *yrkEFHI* operon, contains putative sulfur carriers that are very likely to assist in the cellular response to diamide stress (Fig. 5b and Supplementary Note 6).

Also, the WapA iModulon contains several uncharacterized genes that may be co-regulated by YvrHb, DegU, and WalR and participate in a unique, recently discovered interspecies competition mechanism[72]. This system protrudes fibers from the cell wall to deliver the WapA tRNase to enemy bacteria, potentially compromising cell wall integrity for greater nutrient availability. We observe activation of this iModulon under starvation

conditions and repression under cell wall stress (Fig. 5c), consistent with its putative function.

The other uncharacterized iModulons which are not likely to be noise are prophage elements, whose regulatory mechanisms and effect on phenotype warrant further study. See Supplementary Data 8 for their gene sets, Supplementary Data 9 for summaries of their activating conditions, and Supplementary Data 6 for graphical summaries.

## Discussion

Here, we decomposed the existing, high-quality *B. subtilis* expression data set[5] using ICA. This decomposition identified 83 iModulons in the transcriptome whose overall activity can explain 72% of the variance in gene expression across the wide variety of conditions used to generate the data set. Sixty-six of the iModulons correspond to specific biological functions or transcriptional regulators. We analyzed the gene sets and activity levels of the iModulons and presented findings that either agree with existing knowledge or generate hypotheses that could be tested in future studies. The remaining 17 iModulons are independent signals with no coherent biological meaning.

Through the application of ICA, we were able to identify well-studied gene sets with high accuracy (such as the MalR and FadR iModulons), and uncover insights that suggest candidate underlying mechanisms. We discovered unexpected relationships between stress, metabolism, and lifestyle: ethanol appears to stimulate tryptophan synthesis, histidine utilization may be a feature of quorum sensing, DNA damage may induce swarming, and the iron chelator pulcherrimin could help to signal the stationary phase. The tiered response to phosphate limitation was captured as two separate iModulons, which may provide evidence for variable promoter affinity across the known regulon. ICA accurately decomposed sporulation into a small set of steps which allow sporulation progress to be tracked; this revealed unexplained, unusual activity for SigG in anaerobic conditions. The global transcriptional response to sporulation in metabolism and lifestyle governance was summarized concisely in three stages by iModulon activities. Finally, three iModulons contain mostly uncharacterized gene sets, which represent a promising area for further research. Overall, we have demonstrated that ICA produces biologically relevant iModulons with hypothesis-generating capability from microarray data in this model gram-positive organism.

The iModulon genes and activity profile data (Supplementary Data 3–5), along with graphical summaries (Supplementary Data 6) are available for examination by microbiologists with specific interests about functions in *B. subtilis* that are not detailed in this article. We also have an online resource, imodulondb.org, where users can search and browse all iModulons from this data set and view them with interactive dashboards[26]. Code for our analysis pipeline is maintained on github (https://github.com/SBRG/precise-db). There is a strong potential for protein identification, transcription factor discovery, metabolic network insights, function assignment, and mechanism elucidation derived from this iModulon structure of the TRN.

As with all machine learning approaches, the results from ICA improve as it is provided with more high-quality data[10]. Future research may append unique conditions to this data set and observe the changes to the set of iModulons it finds. Perhaps multi-purpose iModulons will be divided into their biologically accurate building blocks, the noise will be removed, and new regulons will emerge as the signal-to-noise ratio improves. With enough additional data, ICA could potentially characterize the entire TRN in great detail, a goal that has been the subject of research for over half a century. Ultimately, this could be the foundation for a comprehensive, quantitative, irreducible TRN.

## Methods

**Data acquisition and preprocessing**. We obtained normalized, log2-transformed tiling microarray expression values from Nicolas et al.[5] (GEO accession number GSE27219), which span 5875 transcribed regions (4292 coding sequences and 1632 previously unannotated RNAs) and 269 sample profiles (104 conditions). The strain used, BSB1, is a prototrophic derivative of the popular laboratory strain, 168. Three samples (S3_3, G + S_1, and Mt0_2) were removed so that the Pearson $R$ correlation between biological replicates was no <0.9, except in the case of sporulation hour 8, where $n = 2$ and $R = 0.89$ (Supplementary Fig. 1a). To obtain more easily interpretable activity levels, we centered the data by subtracting the mean in the M9 exponential growth condition from all gene values. This is consistent with our prior work in *E. coli*, where a similar condition was chosen for this purpose. All activities are therefore relative to a known, consistent baseline condition.

**Independent component analysis**. Independent component analysis decomposes a transcriptomic matrix ($\mathbf{X}$, Supplementary Data 2) into independent components ($\mathbf{M}$, Supplementary Data 3) and their condition-specific activities ($\mathbf{A}$, Supplementary Data 4):

$$\mathbf{X} = \mathbf{M} * \mathbf{A}. \tag{1}$$

ICA was performed as described in Supplementary Methods. Note that the $\mathbf{M}$ matrix was previously called $\mathbf{S}$[10]; it has been changed to avoid confusion with other nomenclature. See Supplementary Methods.

We normalized each component in the $\mathbf{M}$ matrix such that the maximum absolute gene weight was 1. We performed the inverse normalization on the $\mathbf{A}$ matrix to conserve the same values. Therefore, each unit in $\mathbf{A}$ is equivalent to a unit log change in expression if the iModulon were to contain only one gene.

Thresholds were applied to the columns in the $\mathbf{M}$ matrix to acquire gene sets for each iModulon (Supplementary Methods).

**Regulator enrichment**. Regulon information was obtained from *Subti*Wiki[6]. For each iModulon, we obtained all regulators that regulate any gene in their gene sets. We also used all combinations of regulators, denoted by "+" between regulator names, to capture regulons with more than one regulator. For each of those individual regulators and regulator combinations, we obtained a regulon set, a list of all genes that share that regulation. Next, we computed $p$-values for each regulon's overlap with the iModulon gene set using the two-sided Fisher's exact test (FDR < $10^{-5}$)[73,74]. We also computed F1 scores, which are the harmonic averages of precision and recall.

After the sensitivity analysis (Supplementary Methods) determined the appropriate cutoff, significant enrichments for each iModulon were then manually curated (Supplementary Data 7). In most cases, the most significant enrichment was chosen. Some iModulons appeared to be a combination of two or more significantly enriched regulons, so their assigned regulator was a union of both, denoted by "/" between regulator names.

Our regulator enrichments have very high precision and recall scores, but they have an inherent bias because the threshold for iModulon membership was chosen to maximize them. Our method of selecting the threshold improves with the completeness of the TRN annotations (Supplementary Fig. 2d), and would be ineffective for an organism with a very incomplete TRN. We could work around that limitation with approaches using other gene groupings, such as functional, category, or motif enrichments, or by developing approaches that compare iModulons across organisms, such as comparing iModulon size distributions, or leveraging homology with model organisms.

**Differential activation analysis**. We fit a log-normal distribution to the differences in iModulon activities between biological replicates for each iModulon. For a single comparison, we computed the absolute value of the difference in the mean iModulon activity and compared it against the iModulon's log-normal distribution to determine a $p$-value. We performed this comparison (two-tailed) for a given pair of conditions across all iModulons at once and designated significance as FDR < 0.01.

**Reporting summary**. Further information on research design is available in the Nature Research Reporting Summary linked to this article.

## Data availability

All data generated or analyzed during this study are included in this published article (and its Supplementary Information Files). The original data set is from Nicolas, et al.[5] (GEO accession number GSE27219; Supplementary Data 1 and 2 from http://genome.jouy.inra.fr/basysbio/bsubtranscriptome/). Interactive online dashboards for all iModulons and all data are available at https://imodulondb.org under the data set name "*B. subtilis*".

## Code availability

Code for our analysis pipeline is maintained on GitHub (https://github.com/SBRG/precise-db)[75].

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

## Acknowledgements

We thank Dr. Joe Pogliano, Saugat Poudel, and Eammon Riley for helpful discussions and biological insights. This research used resources of the National Energy Research Scientific Computing Center, a DOE Office of Science User Facility supported by the Office of Science of the U.S. Department of Energy under Contract No. DE-AC02-05CH11231. This work was funded by the Novo Nordisk Foundation Center for Bio-sustainability (Grant Number NNF10CC1016517).

## Author contributions

K.R. analyzed data and drafted the paper; A.V.S. designed research; A.V.S. and B.O.P. provided mentorship and guidance throughout. All participated in writing the paper.

## Competing interests

The authors declare no competing interest.
