## [Peer Review File · Nature Communications]

REVIEWER COMMENTS

Reviewer #1 (Remarks to the Author):

Comments to Author

In this manuscript, the author was based on an unsupervised statistical learning algorithm _Independent Component Analysis (ICA) to analyze Nicolas, et al., generated a transcriptomic microarray dataset of *Bacillus subtilis*. A total of 83 robust i-modulons were obtained, they explain 72% of the variance in gene expression of this dataset, 66/83 i-modulons can correspond to specific biological functions or transcriptional regulators. Author present five unexpected i-modulon activations and hypotheses about their mechanisms, this may have an important guiding role for research in these areas. The main transcription steps of sporulation formation were captured by six i-modules, and the gene sets and regulators of these i-modulons corresponded to the know sporulation process, the sporulation formation process is further divided into three main transcription stages, which will help researchers to accurately study the sporulation process. Finally, author proposes three i-modulons that have no obvious characteristics but are worthy of further study. This study has great potential for studying TRN in *Bacillus subtilis*, and as the available data increases, and as the available data increases, the accuracy of i-modulon classification will increase and the related pathway description will be more comprehensive, this will become an important top-down tool for *Bacillus subtilis* to regulate metabolism . Several issues need to be discussed carefully for accept for publication in this journal.

Specific comments:

1. In method section, author centered the data by subtracting the mean in the M9 exponential growth condition from all gene values, Why choose the "mean in the M9 exponential growth condition", please discuss or explain it.
2. There are some errors in the references, such as: "Tan, I. S. & Ramamurthi, K. S. Spore formation in *Bacillus subtilis*. *Environ Microbiol Rep* 6, 212–225 (2014).", *Bacillus subtilis* need use Italics.
3. In supplementary methods section, different i-modulons K2 cutoff value is different, how to define non-important genes, how many percentage of the original data is considered noise at the selected cutoff value.
4. In result section, "63 i-modulons were successfully mapped to a known regulator", what about the coverage of these i-modulons and regulons, please add it.
5. The Latin literature name of the bacteria in the manuscript is not italicized, please modify it. For example: "We previously applied this approach to a large, high quality *Escherichia coli* RNA-seq compendium and extracted 92 i-modulons, two thirds of which exhibited high overlap with known regulons".

Reviewer #2 (Remarks to the Author):

In this article, Rychel et al use an Independent Component Analysis (ICA) approach to identify groups of co-expressed genes that they call i-modulons in the *B. subtilis* transcriptome. The same group had previously used a similar method in *E. coli*. Here, they rely on a previously published, large, transcriptomic dataset and obtain 83 i-modulons. These i-modulons explain 72% of the variance in gene expression in the dataset. Furthermore, based on the composition of the gene groups, they connected about 3/4 of these i-modulons to known regulators of transcription. This observation allows them to determine the precision and recall parameters of their approach by comparing the composition of the i-modulons to that of the corresponding regulons (as described in the SubtiWiki database). This led them to propose new regulatory hypotheses affecting the metabolism and other well characterized processes in *B. subtilis* (biofilm formation and sporulation). In general, this information is of high interest to the scientific community working with *B. subtilis* and other Gram-positive bacteria; however, some important points listed below

need to be addressed.

1) While the introduction summarizes the advantages of ICA over other TRN inference methods, it does so in a very general way. There is no mention of previous attempts to infer the *B. subtilis* TRN and how the authors' approach differs from those previous attempts. More specifically, I was surprised that the authors omitted to include in their references the article by Arrieta-Ortiz et al. 2015 (PMID: 26577401) describing the *B. subtilis* TRN in a way that is more comprehensive than the current study. It similarly recalled a large number of known regulatory interactions and used two large transcriptomic datasets (the one used in the current study and a second one of equivalent size generated specifically for that study). The Rychel et al approach has the advantage of being unbiased, while the Arrieta-Ortiz et al. approach relied on previous knowledge to estimate transcription factor activities. Nevertheless, both methods have their merits and it would be interesting to compare them more directly. For instance, the Inferelator approach of Arrieta-Ortiz et al. does a better job at identifying genes regulated by the sporulation sigma factors. In addition to that study, other articles that could be mentioned include Fadda et al 2009 (PMID: 20023724) and Leyn et al 2013 (PMID: 23504016).

2) It probably is beyond the scope of this paper to attempt to experimentally validate some of the new predictions. Nevertheless, it would help to discriminate more efficiently between what constitutes a hypothesis and a validated interaction. Subheadings like "DNA damage stimulates swarming", "An iron chelator signals the stationary phase" give the impression that these statements have been validated even though they are only hypotheses that have not yet been experimentally confirmed.

3) Analysis of sporulation i-modulons. The authors acknowledge on p.9 that "The conclusions drawn from these i-modulons are limited by the complexity of sporulation". Even with this limitation in mind, several issues remain:

a) on p.9, last sentence "including identifying 20 unknown proteins that could now be classified as putatively sporulation-related (Supplementary Fig. S7b, Dataset S8)" It is not clear looking at Dataset S8 what these 20 unknown proteins are (they may have to be listed in a separate table). Most of the genes listed in the SigE, SigE/G, SigG, SigK1 and SigK2 i-modulons were also described as sporulation genes in the Arrieta-Ortiz et al. paper, so in that sense they are not novel (even if some of them were not included as sporulation genes in SubtiWiki).

Furthermore, many of these genes were already described in two other publications Eichenberger et al 2004 (PMID: 15383836) and Wang et al 2006 (PMID: 16497325) on the mother cell and forespore lines of gene expression, respectively.

b) on p. 10, the statement "we believe it was not identified because its genes are expressed to a lesser degree in the overall population due to the small size of the forespore at this stage" is highly speculative and probably incorrect. SigG like SigF is active in the forespore as well and one of the i-modulons could be connected to SigG. I believe that there are alternative explanations (not mutually exclusive) that could explain the lack of a SigF i-modulon. In fact, some known SigF-dependent genes appear in the SigE, SigG and SigE/SigG i-modulons. This is because many SigF dependent genes are also under SigG control (their consensus sequences are very similar). Furthermore, because SigF is active in the forespore and SigE in the mother cell, many genes under the control of these regulators are expressed simultaneously and the method does not easily distinguish between the two cell compartments.

c) on p.10 "The SigG i-modulon follows; it contains more coat maturation proteins" This is inaccurate, it is well established that coat proteins are only expressed in the mother cell (under SigE and SigK control) and not in the forespore. SigG only regulates gene expression in the forespore.

d) on p.10 "the SigK regulon is split into two i-modulons with functions including DNA protection,". This is also inaccurate, proteins involved in DNA protection are exclusively found in the forespore (under SigF and SigG control), so they cannot be under SigK control (which is only active in the mother cell).

e) on p.10 "the SigK regulon is split into two i-modulons". This should be explained in more detail. Is it because of the role of the SigK-dependent regulator GerE, which is required for the expression of the late SigK-controlled genes via a feed-forward loop (see Eichenberger et al 2004; PMID: 15383836)?

f) on p.11 “Despite that, recent research into the activation of SigG has stated that it should never be expressed during vegetative growth”, the formulation is awkward and should be rephrased. What is known from previous research is that premature activation of SigG blocks spore formation (Chary et al 2007 PMID: 17921305) and that ectopic activation of SigG is limited by negative feedback loop (Serrano et al 2011; PMID: 21935351). In light of these other publications, the hypothesis that “We therefore propose that there may be an anaerobic mechanism through which SigG or its known targets are activated, distinct from sporulation” does not stand on firm ground.

4) Based on the points raised above, I believe that the sentence “The TRN also facilitates quantification of population-level sporulation states, revealing a putative anaerobic metabolism role for SigG” is an overstatement and should be removed from the abstract unless experimental evidence can be provided in support of the assumption. This also applies to p.14 “this led to the discovery of unusual activity for SigG in anaerobic conditions”. To qualify as a “discovery”, this observation would have to be experimentally validated, for instance by introducing a reporter of SigG activity and examining expression of the reporter under anaerobic conditions.

5) When connecting i-modulons to specific regulators (p.16 “Regulator Enrichment” section, it would help to specify whether the regulator acts as an activator (e.g. sigma factors), a repressor (e.g. CodY, AbrB) or both (e.g. Spo0A, SpoIIID, GerE). A sentence like “The AbrB i-modulon is activated as well” on p. 12 is a bit strange, because AbrB is a repressor and genes under AbrB control are in fact derepressed at this stage.

Reviewer #3 (Remarks to the Author):

On this work, Rychel et al. identify sets of genes that are independently modulated (i-modulons) by using independent component analysis (ICA) on microarray data of *B. subtilis*. The work provides a new application to *B. subtilis* of a methodology previously published by the authors using *E. coli* expression data [PMID: 31797920]. They compute the overlap of the i-modulons with the regulons from the *B. subtilis* regulatory network and found high precision and sensitivity. Finally, they generate several biological hypotheses through an in-deep analysis of the i-modulons found.

Major concerns:

1. The precision and recall scores of the regulons are outstanding when compared with previous efforts for the inference of regulatory networks from transcriptomics [PMID: 22796662]. However, the authors maximize the scores when they perform the “sensitivity analysis”. They select the regulons maximizing the F1 score. Therefore, their precision and recall scores are the highest they can be with the i-modulons obtained. It is useful to formulate new biological hypotheses, as the authors do. However, the analysis is not applicable if a regulatory network (having a good completeness) is not available for the organism of interest. The authors should mention this and discuss alternatives to work around this limitation.
2. In the section Methods/Regulator Enrichment, it is not quite clear the rules followed to select the regulator(s). Namely, I concern about the scenario where a transcription factor (TF) is included in the i-modulon but none of its target genes are in the same i-modulon, the TF is discarded as a potential regulator? If so, there is a problem regarding the incompleteness of the regulatory network used. A poorly studied TF may be a regulator of genes in the i-modulon but none of those targets may have been found, or even worse it could not have been characterized as a TF yet.
3. As a consequence of the previous point, incompleteness is important to the analysis. Recently, incompleteness of regulatory networks has been quantified and data to assess its effect is available [PMID: 32542109]. The authors must carry out a sensitivity analysis and discuss how the incompleteness of the regulatory network affect their analysis and results.
4. Continuing with the point 2, the authors miss several opportunities when selecting the

regulators. For example, they could use motif discovery algorithms to find a common regulatory site in the upstream sequence of the genes in an i-modulon, and compare it with previously characterized binding motifs [PMID: 17324271] to reduce the limitations imposed by the incompleteness of the regulatory network, which might cause “false” positives.

Minor concerns:

1. In previous work with *E. coli*, the authors showed that their pipeline performs better on RNA-seq data than it does with Microarrays data. However, they used Microarray data here. Is it because of a lack of RNA-seq data for *B. subtilis*? If so, it should be mentioned.
2. Since the i-modulons resemble the regulons, do the distribution of their size follow a power law as the connectivity of the TFs in a regulatory network does [PMID: 22728391, 24237659]?
3. How many i-modulons contain only one TF? Do they correlate better with the known regulons? Using the example of the FadR i-modulon and its genes regulated by other TFs, the prediction of the i-modulons is better for genes with a single TF? How is this related to the observations in [PMID: 30462289] (see Fig.2)?
4. The i-modulons classified as “subset” contain global regulators. I expect the global regulators in the TRN of *B. subtilis* to be included in multiple i-modulons [PMID: 24237659]. If so, is there a correlation between the number of i-modulons membership and the number of regulated genes?
5. The “well-matched” i-modulons mostly correspond to local regulators. Perhaps the global regulators might be outscored by a local regulator on the “Sensitivity analysis” with small i-modulons (The i-modulon performs a high precision/recall with the local regulon but high precision and low recall with the global regulon, making the F1-score higher for the local regulator).
6. At the beginning of the section “Results”: “Unlike regulons, which are sets of co-regulated genes based on experimentally confirmed DNA binding sites of transcriptional regulators...” The interactions of a regulon are not always confirmed, and regulatory networks may contain indirect regulations. It depends on the confidence level and curation state of the network used (see evidence classification in RegulonDB and SubtiWiki).
7. I did not understand what is the objective of this sentence: “– we did not provide this untargeted statistical approach with any information about the known regulon structure of the TRN (Fig. 1a).” in the first paragraph of the section “Results”.
8. “The TRN covers 2,235 gene/i-modulon relationships, of which 1,536 are known gene/regulator interactions and 699 are new (Dataset S8).” The TRN is that obtained from SubtiWiki or the i-modulons?
9. What fraction of the *B. subtilis* genome is considered on the transcriptomic dataset? And what fraction is considered on the final i-modulons?
10. The methods section needs to be reviewed for consistency (for example, in the section “I-modulon Threshold Determination” of the supplementary material mention “we computed the top regulator enrichments and F1 scores as described in the following section.” But that method description is in “Regulator Enrichment” in the main text.
11. In the second paragraph of “I-modulon Threshold Determination.” The authors mention seven cases of fine-tuning of the threshold. How do the authors define the non-importance of the genes captured in two of those i-modulons?
12. The work [PMID: 27766091] might be of interest to the authors regarding their hypothesis “DNA damage stimulates swarming”.
13. In Main Fig. 2E, why is the intersection of the PhoP and the SigA regulon in the Venn-diagram instead of only the PhoP regulon?
14. Regarding Main Fig 4, it would also be interesting to see the gene i-modulons activity over the first hour.
15. (Typo). There is a reference to Main Fig. 2H in the description of the Supplementary Fig. S4. I think it was a reference to Main Fig. 2C (as there is no panel H in Main Fig. 2).
16. (Typo??). In the third paragraph of “Independent Component Analysis” in Methods, how is calculated the equivalence to a “unit log(TPM)” using microarrays?

REVIEWER COMMENTS

Reviewer #1 (Remarks to the Author):

Comments to Author

In this manuscript, the author was based on an unsupervised statistical learning algorithm Independent Component Analysis (ICA) to analyze Nicolas, et al., generated a transcriptomic microarray dataset of *Bacillus subtilis*. A total of 83 robust i-modulons were obtained, they explain 72% of the variance in gene expression of this dataset, 66/83 i-modulons can correspond to specific biological functions or transcriptional regulators. Author present five unexpected i-modulon activations and hypotheses about their mechanisms, this may have an important guiding role for research in these areas. The main transcription steps of sporulation formation were captured by six i-modules, and the gene sets and regulators of these i-modulons corresponded to the know sporulation process, the sporulation formation process is further divided into three main transcription stages, which will help researchers to accurately study the sporulation process. Finally, author proposes three i-modulons that have no obvious characteristics but are worthy of further study. This study has great potential for studying TRN in *Bacillus subtilis*, and as the available data increases, and as the available data increases, the accuracy of i-modulon classification will increase and the related pathway description will be more comprehensive, this will become an important top-down tool for *Bacillus subtilis* to regulate metabolism . Several issues need to be discussed carefully for accept for publication in this journal.

Specific comments:

1. In method section, author centered the data by subtracting the mean in the M9 exponential growth condition from all gene values, Why choose the “mean in the M9 exponential growth condition”, please discuss or explain it.

This is consistent with our prior work in *E. coli*, where a similar condition was chosen for this purpose. All activities are therefore relative to a known, consistent baseline condition. This clarification has been added.

Also, when analyzing RNA-seq data, absolute expression values (i.e. log-TPM) do not reflect an intrinsic biological measurement. Instead, we normally report values in log-fold-changes between one condition and a reference. In this same vein, we compare all iModulon activities against a reference condition, since only relative iModulon activities are meaningful.
2. There are some errors in the references, such as: "Tan, I. S. & Ramamurthi, K. S. Spore formation in *Bacillus subtilis*. *Environ Microbiol Rep* 6, 212–225 (2014).", *Bacillus subtilis* need use Italics.

Thank you for pointing this out. We have fixed all italics errors.
3. In supplementary methods section, different i-modulons K2 cutoff value is different, how to define non-important genes, how many percentage of the original data is considered noise at the selected cutoff value.

We added the following paragraph, as well as Supplementary Fig. S8, to explain this reasoning: “The two cases (MalR and Rex) in which the threshold was increased are shown in Supplementary Fig. S8. In these cases, we see the tails of the distribution starting a bit higher than their computed thresholds, and the full tail corresponds to a regulon. For this reason, it was appropriate to raise the thresholds in these cases.”

The explained variance of our decomposition with the thresholds used is 72%, so we consider 28% of the original data to be noise.

4. In result section, “63 i-modulons were successfully mapped to a known regulator”, what about the coverage of these i-modulons and regulons, please add it.

The coverage of the iModulons is 36.25%, this statement has been added to the first paragraph of the results section. The TRN from SubtiWiki has a coverage of 58.1%.

5. The Latin literature name of the bacteria in the manuscript is not italicized, please modify it. For example: “We previously applied this approach to a large, high quality *Escherichia coli* RNA-seq compendium and extracted 92 i-modulons, two thirds of which exhibited high overlap with known regulons”.

We have fixed all italics errors.

Reviewer #2 (Remarks to the Author):

In this article, Rychel et al use an Independent Component Analysis (ICA) approach to identify groups of co-expressed genes that they call i-modulons in the *B. subtilis* transcriptome. The same group had previously used a similar method in *E. coli*. Here, they rely on a previously published, large, transcriptomic dataset and obtain 83 i-modulons. These i-modulons explain 72% of the variance in gene expression in the dataset. Furthermore, based on the composition of the gene groups, they connected about 3/4 of these i-modulons to known regulators of transcription. This observation allows them to determine the precision and recall parameters of their approach by comparing the composition of the i-modulons to that of the corresponding regulons (as described in the SubtiWiki database). This led them to propose new regulatory hypotheses affecting the metabolism and other well characterized processes in *B. subtilis* (biofilm formation and sporulation). In general, this information is of high interest to the scientific community working with *B. subtilis* and other Gram-positive bacteria; however, some important points listed below need to be addressed.

1. While the introduction summarizes the advantages of ICA over other TRN inference methods, it does so in a very general way. There is no mention of previous attempts to infer the *B. subtilis* TRN and how the authors' approach differs from those previous attempts. More specifically, I was surprised that the authors omitted to include in their references the article by Arrieta-Ortiz et al. 2015 (PMID: 26577401) describing the *B. subtilis* TRN in a way that is more comprehensive than the current study. It similarly recalled a large number of known regulatory interactions and used two large transcriptomic datasets (the one used in the current study and a second one of

equivalent size generated specifically for that study). The Rychel et al approach has the advantage of being unbiased, while the Arrieta-Ortiz et al. approach relied on previous knowledge to estimate transcription factor activities. Nevertheless, both methods have their merits and it would be interesting to compare them more directly. For instance, the Inferelator approach of Arrieta-Ortiz et al. does a better job at identifying genes regulated by the sporulation sigma factors. In addition to that study, other articles that could be mentioned include Fadda et al 2009 (PMID: 20023724) and Leyn et al 2013 (PMID: 23504016).

A paragraph (“Without using ICA...”) has been added to the introduction to include these references and describe our approach’s advantage of being unbiased. We agree that a direct comparison would be very interesting, but it is out of the scope of this paper.

2. It probably is beyond the scope of this paper to attempt to experimentally validate some of the new predictions. Nevertheless, it would help to discriminate more efficiently between what constitutes a hypothesis and a validated interaction. Subheadings like “DNA damage stimulates swarming”, “An iron chelator signals the stationary phase” give the impression that these statements have been validated even though they are only hypotheses that have not yet been experimentally confirmed.

The word ‘may’ has been added to each subheading to indicate this.

3. Analysis of sporulation i-modulons. The authors acknowledge on p.9 that “The conclusions drawn from these i-modulons are limited by the complexity of sporulation”. Even with this limitation in mind, several issues remain:
 - a. On p.9, last sentence “including identifying 20 unknown proteins that could now be classified as putatively sporulation-related (Supplementary Fig. S7b, Dataset S8)” It is not clear looking at Dataset S8 what these 20 unknown proteins are (they may have to be listed in a separate table). Most of the genes listed in the SigE, SigE/G, SigG, SigK1 and SigK2 i-modulons were also described as sporulation genes in the Arrieta-Ortiz et al. paper, so in that sense they are not novel (even if some of them were not included as sporulation genes in SubtiWiki). Furthermore, many of these genes were already described in two other publications Eichenberger et al 2004 (PMID: 15383836) and Wang et al 2006 (PMID: 16497325) on the mother cell and forespore lines of gene expression, respectively.

We generated the list of 20 genes and compared them against the three papers you listed; it is in Dataset S11. We also rephrased the sentence to: “including identifying 20 uncharacterized proteins whose annotations did not previously reflect a putative relationship to sporulation.” Of the 20 genes, 11 of them (highlighted in Dataset S11) were not mentioned in any of the three papers and didn’t have *SubtiWiki* annotations reflecting sporulation. The other 9 are still included, with notes about Arrieta-Ortiz’s predictions (which match our own in 2/4 cases) and Eichenberger/Wang’s evidence (which agree with us in all 7 cases with varying degrees of confidence).

- b. On p. 10, the statement “we believe it was not identified because its genes are expressed to a lesser degree in the overall population due to the small size of the forespore at this stage” is highly speculative and probably incorrect. SigG like SigF is active in the forespore as well and one of the i-modulons could be connected to SigG. I believe that there are alternative explanations (not mutually exclusive) that could explain the lack of a SigF i-modulon. In fact, some known SigF-dependent genes appear in the SigE, SigG and SigE/SigG i-modulons. This is because many SigF dependent genes are also under SigG control (their consensus sequences are very similar). Furthermore, because SigF is active in the forespore and SigE in the mother cell, many genes under the control of these regulators are expressed simultaneously and the method does not easily distinguish between the two cell compartments.
- This reasoning makes much more sense. We have replaced the previous explanation with the following: “Notably, SigF is the only absent sigma factor; we believe it was not identified because its genes are expressed simultaneously with the SigE, SigG, and SigE/G iModulons, and because many SigF genes are also under SigG control.”
- c. On p.10 “The SigG i-modulon follows; it contains more coat maturation proteins” This is inaccurate, it is well established that coat proteins are only expressed in the mother cell (under SigE and SigK control) and not in the forespore. SigG only regulates gene expression in the forespore.
- We have removed the reference to coat maturation proteins. They must have been included in that list by accident.
- d. On p.10 “the SigK regulon is split into two i-modulons with functions including DNA protection,”. This is also inaccurate, proteins involved in DNA protection are exclusively found in the forespore (under SigF and SigG control), so they cannot be under SigK control (which is only active in the mother cell).
- We have removed the mention of DNA protection.
- e. On p.10 “the SigK regulon is split into two i-modulons”. This should be explained in more detail. Is it because of the role of the SigK-dependent regulator GerE, which is required for the expression of the late SigK-controlled genes via a feed-forward loop (see Eichenberger et al 2004; PMID: 15383836)?
- Fantastic observation, you are correct. We added the following: “The difference between the two SigK iModulons may partially be explained by the action of the TF GerE, which represses members of SigK-1 and activates a large fraction of SigK-2 (Supplementary Fig. S7c-d). This is consistent with the known temporal regulation of the SigK regulon.”, including a citation to Eichenberger. As mentioned, we added a supplementary figure with pie charts of GerE’s annotated effect on each gene in the two iModulons, showing that SigK-1 is repressed by GerE while SigK-2 is not.
- f. On p.11 “Despite that, recent research into the activation of SigG has stated that it should never be expressed during vegetative growth”, the formulation is awkward and should be rephrased. What is known from previous research is that

premature activation of SigG blocks spore formation (Chary et al 2007 PMID: 17921305) and that ectopic activation of SigG is limited by negative feedback loop (Serrano et al 2011; PMID: 21935351). In light of these other publications, the hypothesis that “We therefore propose that there may be an anaerobic mechanism through which SigG or its known targets are activated, distinct from sporulation” does not stand on firm ground.

We agree that this hypothesis was a stretch. We have rephrased the end of this paragraph, included the points you made, and replaced our initial claim with: “We therefore propose further experiments to determine the role of SigG-dependent genes in anaerobiosis.”

4. Based on the points raised above, I believe that the sentence “The TRN also facilitates quantification of population-level sporulation states, revealing a putative anaerobic metabolism role for SigG” is an overstatement and should be removed from the abstract unless experimental evidence can be provided in support of the assumption. This also applies to p.14 “this led to the discovery of unusual activity for SigG in anaerobic conditions”. To qualify as a “discovery”, this observation would have to be experimentally validated, for instance by introducing a reporter of SigG activity and examining expression of the reporter under anaerobic conditions.

We removed “revealing a putative anaerobic metabolism role for SigG” from the abstract. We also rephrased the discussion to state simply “this revealed unexplained, unusual activity for SigG in anaerobic conditions.”

5. When connecting i-modulons to specific regulators (p.16 “Regulator Enrichment” section, it would help to specify whether the regulator acts as an activator (e.g. sigma factors), a repressor (e.g. CodY, AbrB) or both (e.g. Spo0A, SpoIIID, GerE). A sentence like “The AbrB i-modulon is activated as well” on p. 12 is a bit strange, because AbrB is a repressor and genes under AbrB control are in fact derepressed at this stage.

We replaced “activated” with “derepressed” and added the column “Mode of Regulation” to SI Dataset S7, which covers this. We used the specific gene/TF annotations from SubtiWiki, which categorizes sigma factors separately from activation. Though Spo0A is both a repressor and an activator, it activates all of the genes in the Spo0A iModulon, so it is listed as an activator. We use the ambiguous term “regulation” in cases where the genes are both activated and repressed within the iModulon.

Reviewer #3 (Remarks to the Author):

On this work, Rychel et al. identify sets of genes that are independently modulated (i-modulons) by using independent component analysis (ICA) on microarray data of *B. subtilis*. The work provides a new application to *B. subtilis* of a methodology previously published by the authors using *E. coli* expression data [PMID: 31797920]. They compute the overlap of the i-modulons with the regulons from the *B. subtilis* regulatory network and found high precision and

sensitivity. Finally, they generate several biological hypotheses through an in-deep analysis of the i-modulons found.

Major concerns:

1. The precision and recall scores of the regulons are outstanding when compared with previous efforts for the inference of regulatory networks from transcriptomics [PMID: 22796662]. However, the authors maximize the scores when they perform the “sensitivity analysis”. They select the regulons maximizing the F1 score. Therefore, their precision and recall scores are the highest they can be with the i-modulons obtained. It is useful to formulate new biological hypotheses, as the authors do. However, the analysis is not applicable if a regulatory network (having a good completeness) is not available for the organism of interest. The authors should mention this and discuss alternatives to work around this limitation.

The following discussion has been added to the methods “Regulator Enrichment” section: “Our regulator enrichments have very high precision and recall scores, but they have an inherent bias because the threshold for iModulon membership was chosen to maximize them. Our method of selecting the threshold improves with the completeness of the TRN annotations (Supplementary Fig. S2D), and would be ineffective for an organism with a very incomplete TRN. We could work around that limitation with approaches using other gene groupings, such as functional, category, or motif enrichments, or by developing approaches that compare iModulons across organisms, such as comparing iModulon size distributions, or leveraging homology with model organisms.”

2. In the section Methods/Regulator Enrichment, it is not quite clear the rules followed to select the regulator(s). Namely, I concern about the scenario where a transcription factor (TF) is included in the i-modulon but none of its target genes are in the same i-modulon, the TF is discarded as a potential regulator? If so, there is a problem regarding the incompleteness of the regulatory network used. A poorly studied TF may be a regulator of genes in the i-modulon but none of those targets may have been found, or even worse it could not have been characterized as a TF yet.

You are correct that if a TF is in an iModulon but its target genes are not, it is typically not considered to be a potential regulator. This is advantageous in many cases. For example, sigE and sigG are both members of the Spo0A iModulon, as they are targets of Spo0A, each regulating their own iModulon downstream of Spo0A. Through post-transcriptional mechanisms, the expression of their encoding RNA does not correlate with the expression of their target genes, so they are not recognized as part of the same signal in the transcriptome. However, in the case that no significant regulator is enriched for an iModulon, we may look for potential TFs within the iModulon itself. It would be most likely to be present in the iModulon if it also regulates itself in a positive feedback loop. If any of the unknown genes in our 3 ‘new regulon’ iModulons had been transcription factors, we would have predicted this possible mechanism.

In the case of a more incomplete TRN, there would be more cases for which we would need to manually curate TRN assignments, so looking within the iModulon itself would happen more frequently than it does for *Bacillus*.

3. As a consequence of the previous point, incompleteness is important to the analysis. Recently, incompleteness of regulatory networks has been quantified and data to assess its effect is available [PMID: 32542109]. The authors must carry out a sensitivity analysis and discuss how the incompleteness of the regulatory network affect their analysis and results.

We used all legacy TRNs from [PMID: 32542109] to compute average precision and recall scores across the regulated iModulons. Results are shown in Supplementary Fig. S2D. As the TRN becomes more complete over the years, both precision and recall increase. We added a supplementary methods section (Transcriptional Regulatory Network Annotation) and supplementary results section (Precision and recall improve...) to discuss this further.

4. Continuing with the point 2, the authors miss several opportunities when selecting the regulators. For example, they could use motif discovery algorithms to find a common regulatory site in the upstream sequence of the genes in an i-modulon, and compare it with previously characterized binding motifs [PMID: 17324271] to reduce the limitations imposed by the incompleteness of the regulatory network, which might cause “false” positives.

We used a motif discovery algorithm and compared against the previously characterized binding motifs in the PRODORIC database. 29 motifs were identified and 13 of them matched a characterized motif. The results are given in the SI, including a paragraph of supplemental results, a table (SI Dataset S10), and graphical representations in the bottom right corner of the pages in Dataset S6. The results are limited because motifs cannot be confidently found for an iModulon containing less than 4 operons, and because the databases available do not have great coverage of the *Bacillus* transcription factors.

Minor concerns:

1. In previous work with *E. coli*, the authors showed that their pipeline performs better on RNA-seq data than it does with Microarrays data. However, they used Microarray data here. Is it because of a lack of RNA-seq data for *B. subtilis*? If so, it should be mentioned.

Though RNA-seq data exists for *B. subtilis*, the Nicolas, et al. dataset has a comparatively wider diversity of conditions and a more established reputation for data quality. We have shown that the condition space is more important than the technology used [<https://www.biorxiv.org/content/10.1101/2020.04.26.061978v1>], which makes this a good choice of dataset. This explanation has been added to the introduction.

2. Since the i-modulons resemble the regulons, do the distribution of their size follow a power law as the connectivity of the TFs in a regulatory network does [PMID: 22728391, 24237659]?

Yes, their distributions do follow a power law, with $P(k) \sim k^{-0.77}$ and $R^2 = 0.70$. This has been added as supplementary figure S2A-B and sentence in the first paragraph of the results: "The distribution of the number of genes in each iModulon follows a power law, similar to the power law for the connectivity of TFs in literature regulatory networks."

3. How many i-modulons contain only one TF? Do they correlate better with the known regulons? Using the example of the FadR i-modulon and its genes regulated by other TFs, the prediction of the i-modulons is better for genes with a single TF? How is this related to the observations in [PMID: 30462289] (see Fig.2)?

If by "contain" in your first sentence, you mean "are regulated by", then there are 40 iModulons regulated by only one TF and 23 iModulons with more than one TF. The mean precision of the enrichment for single TF iModulons is 83.5%, compared to 73.3% for those with more TFs (Calculated from data in SI dataset S7). In that way, one could claim that single TF iModulons and single TF regulons are better-matched than iModulon/regulon pairs with multiple regulators.

In [PMID: 30462289] Fig. 2, Larsen et al. are concerned with the correlation in expression between TFs and their targets. Their paper highlights the importance of post-transcriptional modifications, time delays, and other confounding variables. We performed a similar analysis using the R^2 coefficient of determination and presented the result in SI Fig. S6C, which builds on our existing analysis of the correlation between TF expression and the iModulons they regulate (SI Results). Similar to Larsen, we obtain a broad range of correlation values. The highest correlations are for TFs that participate in feed-forward loops or are sigma factors lacking post-transcriptional regulation, which matches our expectation. Both of those groups are subsets of the single-TF iModulons, but the single-TF iModulons as a whole do not all exhibit high correlations, since there are also several instances of the pattern shown for MalR.

4. The i-modulons classified as "subset" contain global regulators. I expect the global regulators in the TRN of *B. subtilis* to be included in multiple i-modulons [PMID: 24237659]. If so, is there a correlation between the number of i-modulons membership and the number of regulated genes?

For the subset iModulons, there is a slight correlation ($R^2 = 0.34$) between the number of iModulon genes and the number of genes in the associated regulon. The slope of the best fit line is 3.5, indicating that iModulons typically capture approximately 28.5% of a global regulator's regulon. In cases such as Spo0A, a relatively small subset is identified, whereas cases like SigD include a larger fraction of the regulon. The reason for this difference may be: (1) very strong binding of the master regulator on a smaller subset of genes in cases like Spo0A, which is identified by iModulons while weaker binding and subtle modulation is not captured, (2) the condition space provides better coverage of stress

responses and motility than it does sporulation. We added supplementary figure S2C to address this interesting relationship.

5. The “well-matched” i-modulons mostly correspond to local regulators. Perhaps the global regulators might be outscored by a local regulator on the “Sensitivity analysis” with small i-modulons (The i-modulon performs a high precision/recall with the local regulon but high precision and low recall with the global regulon, making the F1-score higher for the local regulator).

This is correct. We included the term “local regulators” in the statement introducing the well-matched group.

6. At the beginning of the section “Results”: “Unlike regulons, which are sets of co-regulated genes based on experimentally confirmed DNA binding sites of transcriptional regulators...” The interactions of a regulon are not always confirmed, and regulatory networks may contain indirect regulations. It depends on the confidence level and curation state of the network used (see evidence classification in RegulonDB and SubtiWiki).

We have modified this sentence to say that regulons are defined by “a variety of experimental results in literature”.

7. I did not understand what is the objective of this sentence: “– we did not provide this untargeted statistical approach with any information about the known regulon structure of the TRN (Fig. 1a).” in the first paragraph of the section “Results”.

It has been removed and rephrased. The intention was to describe the lack of bias in our approach, which makes the recapitulation of the known structure an exciting result.

8. “The TRN covers 2,235 gene/i-modulon relationships, of which 1,536 are known gene/regulator interactions and 699 are new (Dataset S8).” The TRN is that obtained from SubtiWiki or the i-modulons?

This refers to the iModulon-derived TRN, which has been clarified in the sentence.

9. What fraction of the *B. subtilis* genome is considered on the transcriptomic dataset? And what fraction is considered on the final i-modulons?

Supplementary figure S1D has been added to address this. 95.75% of the features of the genome are considered in the dataset, and 36.25% of the features of the genome are in iModulons. Here, ‘features’ are those included *SubtiWiki*’s gene list, which includes rRNAs, tRNAs, and miscellaneous RNAs.

10. The methods section needs to be reviewed for consistency (for example, in the section “I-modulon Threshold Determination” of the supplementary material mention “we computed the top regulator enrichments and F1 scores as described in the following section.” But that method description is in “Regulator Enrichment” in the main text.

This has been revised to refer to the appropriate section.

11. In the second paragraph of “I-modulon Threshold Determination.” The authors mention seven cases of fine-tuning of the threshold. How do the authors define the non-importance of the genes captured in two of those i-modulons?

We added the following paragraph, as well as Supplementary Fig. S8, to explain this reasoning: “The two cases (MalR and Rex) in which the threshold was increased are shown in Supplementary Fig. S8. In these cases, we see the tails of the distribution starting a bit higher than their computed thresholds, and the full tail corresponds to a regulon. For this reason, it was appropriate to raise the thresholds in these cases.”

12. The work [PMID: 27766091] might be of interest to the authors regarding their hypothesis “DNA damage stimulates swarming”.

Thank you for this reference. We added the sentence: “In addition, this connection may be mediated by interactions between RecA and CheW, which have been observed in *Salmonella enterica*.” This is a very interesting potential mechanism.

13. In Main Fig. 2E, why is the intersection of the PhoP and the SigA regulon in the Venn-diagram instead of only the PhoP regulon?

The intersection of both regulons more accurately describes the iModulon members, since none of the twenty genes that are regulated by PhoP but not SigA are involved in the iModulon. This combination was chosen automatically by the process described in Methods: Regulator Enrichment since it has a higher F1 score than PhoP alone. We dropped “+SigA” from the names for simplicity. A note has been added in the figure caption.

14. Regarding Main Fig 4, it would also be interesting to see the gene i-modulons activity over the first hour.

We agree that early sporulation dynamics would add a fascinating dimension to this analysis. Unfortunately, the sporulation time course is provided in 1 hour intervals, so we do not have access to data that can expose these dynamics.

15. (Typo). There is a reference to Main Fig. 2H in the description of the Supplementary Fig. S4. I think it was a reference to Main Fig. 2C (as there is no panel H in Main Fig. 2).

This reference, along with another reference in the same figure caption, have been corrected.

16. (Typo??). In the third paragraph of “Independent Component Analysis” in Methods, how is calculated the equivalence to a “unit log(TPM)” using microarrays?

You are correct that this is a typo. We removed the “(TPM)” part of this statement. There is a log change in expression, but microarrays do not use TPM.

Additional Changes Made By the Authors

1. We have renamed “i-modulons” to “iModulons” and reflected that change throughout the documents.
2. We built a website (imodulondb.org) which features this and two related works, and allows users to browse interactive iModulon dashboards similar to SI Data S6. We have cited the upcoming Nucleic Acids Research article about this database and included the link in the abstract and data availability section.

REVIEWERS' COMMENTS

Reviewer #2 (Remarks to the Author):

The authors have done an excellent job revising the manuscript and addressing the reviewers' suggestions.

Reviewer #3 (Remarks to the Author):

All my concerns have been properly addressed by the authors, and the manuscript is substantially improved.